# Control of Tooth Form Deformation in Heat Treatment of Spiral Bevel Gears Based on Reverse Adjustment of Cutting Parameters

**DOI:** 10.3390/ma16114183

**Published:** 2023-06-04

**Authors:** Ganhua Liu, Xiaodong Huo, Shiyi Deng

**Affiliations:** School of Mechanical and Electrical Engineering, Jiangxi University of Science and Technology, Ganzhou 341400, China; huoxiaodong1997@163.com (X.H.); dengshiyi1998@163.com (S.D.)

**Keywords:** spiral bevel gear, heat treatment, tooth form error, cutting parameter, reverse engineering

## Abstract

The tooth surface structure of spiral bevel gear is complex and requires high machining accuracy. In order to reduce the tooth form deformation of heat treatment, this paper proposes a reverse adjustment correction model of tooth cutting for heat treatment tooth form deformation of spiral bevel gear. Based on the Levenberg–Marquardat method, a stable and accurate numerical solution for the reverse adjustment amount of the cutting parameters is solved. Firstly, a mathematical model of the tooth surface of spiral bevel gears was established based on the cutting parameters. Secondly, the effect law of each cutting parameter on tooth form was studied by using the method of small variable perturbation. Finally, based on the tooth form error sensitivity coefficient matrix, a reverse adjustment correction model of tooth cutting is established to compensate the heat treatment tooth form deformation by reserving the tooth cutting allowance in the tooth cutting stage. The effectiveness of the reverse adjustment correction model of tooth cutting was verified through experiments on reverse adjustment of tooth cutting processing. The experimental results show that the accumulative tooth form error of the spiral bevel gear after heat treatment is 199.8 μm, which is reduced by 67.71%, and the maximum tooth form error is 8.7 μm, which is reduced by 74.75%, after reverse adjustment of cutting parameters. This research can provide technical support and a theoretical reference for heat treatment tooth form deformation control and high-precision tooth cutting processing of spiral bevel gears.

## 1. Introduction

Spiral bevel gear, as an important transmission part of intersecting shafts, is widely used in automobile, aviation, mining and other fields for its advantages of smooth transmission, strong bearing capacity and low noise [1,2,3]. The high-precision machining of the tooth surface of spiral bevel gears is the basis for ensuring their excellent working performance [4,5]. The heat treatment process of spiral bevel gears, under the interaction of various factors such as material composition, internal stress, and temperature changes inevitably leads to tooth form deformation, resulting in tooth form errors that directly affect the meshing transmission performance and reduce the working life of the gears [6,7,8,9,10]. Therefore, it is necessary to control the heat treatment tooth form deformation of spiral bevel gears.

Gear heat treatment deformation has always been an urgent problem to be solved in production and processing. Faced with the problem of heat treatment deformation, many scholars have proposed control measures from different perspectives. Dybowski uses high-pressure gas quenching instead of traditional oil quenching to reduce gear quenching deformation for AMS6265 steel aviation gears [11]. Zhang added the microalloying element Nb during the high-temperature carburization stage of low-carbon alloy steel gears, which refined the austenite grains and reduced the deformation during the gear heat treatment process [12]. Kim found through thermodynamic coupling simulation that using induction heating instead of traditional furnace heating to heat the gears to the carburizing temperature can effectively reduce the heat treatment deformation of the gears [13]. The above methods can effectively reduce the heat treatment deformation of gears, but due to the heat treatment processing method, the cost is high and cannot be used for industrial mass production. Wang optimized the heat treatment process parameters of 12Cr2Ni4 steel gears to reduce tooth form deformation in heat treatment [14]. Li optimized the process parameters of the high-pressure gas quenching process of 5130 carburizing steel tooth ring and reduced the deformation of heat treatment [15]. By optimizing heat treatment process parameters, heat treatment deformation can be reduced to a certain extent, but at the same time, gear hardness and fatigue resistance often decrease, and there is no good balance between deformation and performance.

Sun summarized and analyzed the deformation law of the cylindrical gear tooth form of automotive transmissions by using the method of reverse correction in the shaving stage; the heat treatment deformation was effectively offset and the tooth surface accuracy was improved. At the same time, the heat treatment performance such as tooth surface hardness and fatigue resistance were not affected [16]. For spiral bevel gears produced in large quantities using the same billet, the tooth form deformation in heat treatment is regular. After calculating the deformation amount, machining allowance can be reserved during the cutting stage to compensate for the tooth form deformation in heat treatment, without affecting the heat treatment performance of the gears. The tooth surface of spiral bevel gears is a complex spatial surface. How to reserve appropriate machining allowance on the tooth surface by adjusting the cutting parameters is a key and difficult research point. Artoni puts forward a compensation and correction method of tooth form error in the grinding stage of spiral bevel gear and quasi-hyperboloid gear [17]. Mu proposed an innovative high-order tooth surface modification method to reduce the operating vibration of high degree-of-fit spiral bevel gear transmission [18]. Xiang proposed a system geometric error analysis and compensation method for a six-axis spiral bevel gear grinding machine with real-time error compensation [19]. Peng proposed a constant value compensation method for collaborative machine tools considering both the geometric and physical properties of the quasi-hyperboloid gear tooth surface [20]. Lei established a non-orthogonal spiral bevel gear grinding correction model with the optimization objectives of tooth surface geometric accuracy and transmission performance [21]. In the tooth surface grinding stage, the processing parameter reverse correction method used to compensate the tooth form error is also suitable for the tooth cutting stage. However, the above studies, when solving the problem of processing parameter reverse, chose to convert it into a linear least square method for approximate solution and ignore the ill-condition or singular problems of Jacobian matrix caused by nonlinear characteristics, and often failed to obtain stable numerical solutions. Ding, based on the application of nonlinear analysis and calculation algorithm, proposed a fixed value correction method for a high-precision grinder to realize the machining of high-precision quasi-hyperboloid gear [22]. Wan established a collaborative manufacturing system model that comprehensively considers the accuracy of tooth surface geometry and contact performance, and obtained stable numerical solutions by using the Levenberg–Marquardt method [23]. The above research provides a reference for compensating for tooth form deformation in heat treatment by reserving machining allowance during the tooth cutting stage by adjusting the cutting parameters.

In recent years, with the development of computer technology, more and more people have adopted numerical simulation technology to study the heat treatment process of mechanical parts. Compared with actual heat treatment processing research, numerical simulation of heat treatment has advantages such as low cost, high efficiency and full process visualization. Kim considered diffusion phase transition and displacement phase transition kinetics during finite element simulation of carburization heat treatment for automotive gear rings, and predicted the heat treatment deformation of the gear rings [24]. Bouissa and Bohlooli studied the deformation and residual stress distribution of large-sized forged steel blocks during water quenching using a finite element model [25]. Esfahani et al. used finite element method to numerically simulate the quenching process of low alloy steel gears, and studied the stress changes and gear volume changes during the phase transformation process of gears in two different quenching media: oil and water [26]. Due to the complex structure of spiral bevel gears, there is still limited research on numerical simulation of heat treatment of spiral bevel gears. In this paper, the visualization of the heat treatment process of spiral bevel gear is realized by multi-field coupling heat treatment simulation.

In summary, this article proposes a reverse adjustment correction model of tooth cutting for heat treatment tooth form deformation of spiral bevel gear, which addresses the problem of tooth form deformation in heat treatment. The tooth form correction method is applied to the control of tooth form deformation in heat treatment of spiral bevel gears. Firstly, a mathematical model of the tooth surface of spiral bevel gears was established based on the cutting parameters. Secondly, the influence of each cutting parameters on tooth form error was analyzed through the method of artificially assigning small variable perturbations to the cutting parameters. Finally, based on the sensitivity coefficient matrix of tooth form error, a reverse adjustment correction model of tooth cutting for heat treatment tooth form deformation of spiral bevel gear was established using the Levenberg–Marquardt method to obtain a stable numerical solution for the reverse adjustment amount of tooth cutting parameters. By adjusting the cutting parameters, machining allowance was reserved during the cutting stage to compensate for the heat treatment tooth form deformation, and the control of the heat treatment tooth form deformation of spiral bevel gears was achieved.

## 2. Establishment of Mathematical Model for Spiral Bevel Gear Tooth Surface

### 2.1. Establishment of Theoretical Tooth Surface Equation

The form of the tooth surface during the machining of spiral bevel gears is determined by the cutting tool parameters and machine tool processing parameters (i.e., cutting parameters). The mathematical modeling of the tooth surface of spiral bevel gears based on cutting machining parameters is the process of establishing the mapping relationship between cutting machining parameters and tooth surface shape. Based on the meshing principle of gear and the cutting principle of spiral bevel gear, combined with the kinematical relationship between the cutting tool, the machine tool components and the wheel blank, the equation of tooth surface can be derived from the blade equation through the coordinate transformation [27,28]. In this paper, the machining method of small spiral bevel gear is the tool inclination method, and the machining method of large spiral bevel gear is the generating method, taking the establishment of the equation of pinion gear tooth surface as an example.

When the cutting disc rotates around its central axis for tooth cutting, the blade forms a conical generating surface. The blade equation and its unit normal equation [29] can be expressed as:(1)$$r_{t}\left(s_{1},\theta_{1}\right)=\left[\begin{array}{c} \left(r_{01}\pm s_{1}\cdot \sin\alpha_{1}\right)\cos\theta_{1}\\ \left(r_{01}\pm s_{1}\cdot \sin\alpha_{1}\right)\sin\theta_{1}\\ -s_{1}\cdot \cos\alpha_{1}\\ 1 \end{array}\right]$$
(2)$$n_{t}=\left[\begin{array}{c} \cos\alpha_{1}\cos\theta_{1}\\ \cos\alpha_{1}\sin\theta_{1}\\ \pm \sin\alpha_{1}\\ 1 \end{array}\right]$$
where $r_{01}$ is the point radius of blade, $s_{1}$ is the distance from any point on the blade to the cutter point, $\alpha_{1}$ is the pressure angle of the blade, $\theta_{1}$ is the blade phase angle. The “−” and “+” correspond to the inner and outer blades, respectively.

Figure 1a shows the structure diagram of a spiral bevel gear machining machine tool with blade tilt drum.

According to the structure of the machine tool, the coordinate system of the motion relationship between the cutting tool, various components of the machine tool and the workpiece can be established, as shown in Figure 1b. $S_{t}=\left\{O_{t};x_{t},y_{t},z_{t}\right\}$ is the cutter coordinate system, $S_{b}=\left\{O_{b};x_{b},y_{b},z_{b}\right\}$ is the coordinate system of the blade tilt drum, $S_{e}=\left\{O_{e};x_{e},y_{e},z_{e}\right\}$ is the coordinate system of the blade swivel drum, $S_{c}=\left\{O_{c};x_{c},y_{c},z_{c}\right\}$ is the coordinate system of the cradle, $S_{o}=\left\{O_{o};x_{o},y_{o},z_{o}\right\}$ is the coordinate system of the machine tool, $S_{n}=\left\{O_{n};x_{n},y_{n},z_{n}\right\}$ is the coordinate system of transition, $S_{q}=\left\{O_{q};x_{q},y_{q},z_{q}\right\}$ is the coordinate system of workpiece and $S_{p}=\left\{O_{p};x_{p},y_{p},z_{p}\right\}$ is the coordinate system of rotating workpiece.

$i,j,E_{01},q_{1},X_{B1},S_{1},X_{1},\delta_{M1}$ are, respectively, the blade tilt angle, blade swivel angle, vertical offset, basic cradle angle, sliding base feed setting, cutter radial setting, increment of machine center to back, and machine root angle during tooth cutting processing, $\varphi_{1}$ is the gear rotation angle during gear processing, $\varphi_{c}$ is the cradle rotation angle at the corresponding time; roll ratio $i_{01}$ characterizes the relative motion relationship between the rotational speed of the cradle and the rotational speed of the workpiece.

Through coordinate matrix transformation, the cutter blade equation $r_{t}\left(s_{1},\theta_{1}\right)$ is transformed from the tool coordinate system to the coordinate system of the rotating workpiece, and the cutter blade equation $r_{t}^{\left(p\right)}\left(s_{1},\theta_{1},\varphi_{1}\right)$ in the coordinate system of rotating workpiece is obtained. Each transformation matrix can be obtained by the coordinate transformation method of multi-body kinematics [29,30]. According to the gear meshing principle [31], the simultaneous meshing equation $f\left(s_{1},\theta_{1},\varphi_{1}\right)=0$, the equations are set up as follows:(3)$$\left\{\begin{array}{l} r_{t}^{\left(p\right)}\left(s_{1},\theta_{1},\varphi_{1}\right)=M_{pq}(\varphi_{1})M_{qn}(X_{1},\delta_{M1})M_{no}(E_{01},X_{B1})M_{oc}(\varphi_{c})M_{ce}(q_{1},S_{1})M_{eb}(j)M_{bt}(i)r_{t}\left(s_{1},\theta_{1}\right)\\ f\left(s_{1},\theta_{1},\varphi_{1}\right)=n\cdot v_{12}=0 \end{array}\right.$$
where **n** is the expression of the cutter blade unit normal vector in t the coordinate system of rotating workpiece; $v_{12}$ is the relative speed of the tooth flank and the cutter blade at the cutting point. Solve equations to eliminate variable $s_{1}$; the theoretical tooth surface equation $r_{1}\left(\theta_{1},\varphi_{1},k\right)$ of the pinion can be obtained, where k characterizes the cutting parameters of the theoretical tooth surface and can be expressed as $k=\left\{i,j,\delta_{M1},E_{01},q_{1},S_{1},X_{B1},X_{1},i_{01},\alpha_{1},r_{01}\right\}$.

### 2.2. Establishment of Gear Three-Dimensional Model

Table 1 shows the basic design parameters of the pinion, and Table 2 shows the cutting machining parameters of the pinion.

According to the pinion tooth surface equation and gear parameters, the array calculation function and graph rendering function of MATLAB are used to solve and draw the discrete points of the tooth surface, as shown in Figure 2a.

The discrete points of the tooth surface were imported into the UG three-dimensional (3D) modeling software, and the reverse modeling from the discrete points to the tooth surface was carried out. Then, the 3D model of the pinion was constructed through array, Boolean operation and other commands, as shown in Figure 2b.

## 3. Simulation of Spiral Bevel Gear Multi-Field Coupling Heat Treatment

### 3.1. Material and Heat Treatment Process

The spiral bevel gear researched in this paper is made of low carbon alloy steel 20CrMnTi, and its material chemical elements are shown in Table 3.

The thermal and physical property parameters, mechanical property parameters and phase transition parameters of 20CrMnTi material in heat treatment were calculated by JMatPro, a professional material property calculation software, and imported into DEFORM software.

In this paper, the heat treatment type of spiral bevel gears is carburizing—quenching—tempering heat treatment. Its main processes are: degreasing, heating, carburizing, quenching and low-temperature tempering. Carburizing is divided into two stages: strong carburizing and diffusion. The heat treatment process flow is shown in Figure 3.

### 3.2. Finite Element Mesh Generation

The heat treatment geometric model is a spiral bevel gear pinion shown in Figure 4. Since spiral bevel gear is a symmetrical geometry about the central axis, in order to improve the calculation efficiency, the gear was cut and 1/3 of the gear was taken for simulation. The finite element mesh model is shown in Figure 4. The mesh type is tetrahedral mesh with 36,689 nodes and 170,070 elements.

### 3.3. Heat Treatment Simulation Results

After heat treatment, the carbon element on the tooth surface of spiral bevel gear is evenly distributed, the carbon content on the tooth surface is about 0.754 and the depth of the carburized layer is 1.05 mm, as shown in Figure 5a.

The martensite content of the tooth surface is 0.988, and the martensite content of the gear core is 0.362, as shown in Figure 5b. The hardness of the tooth surface is 62.8HRC, and the hardness of the tooth core is 33HRC, which meets the performance requirements of hardness and wear-resistant surface and strong and tough core of spiral bevel gear, as shown in Figure 5c. However, deformation is inevitable in the process of heat treatment. The deformation of the tooth surface in heat treatment is shown in Figure 5d. The tooth surface error caused by heat treatment deformation is 0.01527~0.089608 mm.

## 4. Research on the Influence of Cutting Parameters on the Tooth Form Error

In order to study the influence of each cutting parameter on the tooth form structure, the adjusted tooth surface equation after adjusting the cutting parameters is deduced based on the theoretical tooth surface equation. Observe the influence of each cutting parameter’s fine adjustment on the tooth form error by using the method of artificially given small variable perturbation, and analyze the influence weight on the tooth form error of each order.

### 4.1. Establishment of Equation for Adjusted Tooth Surface

After adjustment, the cutting parameters will deviate from the theoretical design value, where $k^{\prime }=\left\{i^{\prime },j^{\prime },{\delta^{\prime }}_{M1},{E^{\prime }}_{01},q^{\prime },{S^{\prime }}_{1},{X^{\prime }}_{B1},{X^{\prime }}_{1},\;{i^{\prime }}_{01},{\alpha^{\prime }}_{1},{r^{\prime }}_{01}\right\}$ characterizes the cutting parameters of the adjusted tooth surface, and where $i^{\prime },j^{\prime },{\delta^{\prime }}_{M1},{E^{\prime }}_{01},q^{\prime },{S^{\prime }}_{1},{X^{\prime }}_{B1},{X^{\prime }}_{1},\;{i^{\prime }}_{01},{\alpha^{\prime }}_{1},{r^{\prime }}_{01}$ are the adjusted blade tilt angle, blade swivel angle, machine root angle during tooth cutting processing, vertical offset, basic cradle angle, cutter radial setting, sliding base feed setting, increment of machine center to back, roll ratio, pressure angle of the blade and radius of the blade point, which can be expressed by the following equation:(4)$$\left\{\begin{array}{l} i^{\prime }=i+\Delta i,\;j^{\prime }=j+\Delta j,\;{q^{\prime }}_{1}=q_{1}+\Delta q_{1}\\ {\delta^{\prime }}_{M1}=\delta_{M1}+\Delta \delta_{M1},\;{E^{\prime }}_{01}=E_{01}+\Delta E_{01}\\ {S^{\prime }}_{1}=S_{1}+\Delta S_{1},\;{X^{\prime }}_{B1}=X_{B1}+\Delta X_{B1}\\ {X^{\prime }}_{1}=X_{1}+\Delta X_{1},\;{i^{\prime }}_{01}=i_{01}+\Delta i_{01}\\ {\alpha^{\prime }}_{1}=\alpha_{1}+\Delta \alpha_{1},\;{r^{\prime }}_{01}=r_{01}+\Delta r_{01} \end{array}\right.$$
where $\Delta i,\Delta j,\Delta q_{1},\Delta \delta_{M1},\Delta E_{01},\Delta S_{1},\Delta X_{B1},\Delta X_{1},\Delta i_{01},\Delta \alpha_{1},\Delta r_{01}$ are the adjustment amounts of each tooth cutting parameter. After the machining parameters are adjusted, only the relevant cutting machining parameters involved in the tooth surface equation change and the equation form remains unchanged. Therefore, the equation of adjusting tooth surface ${r^{\prime }}_{1}\left(\theta_{1},\varphi_{1},k^{\prime }\right)$ can be derived according to the theoretical tooth surface equation.

### 4.2. Calculation of Tooth Form Error

The tooth surface error of spiral bevel gear can be divided into tooth pitch deviation and tooth form error; the tooth surface of spiral bevel gear is rotationally symmetrical about the central axis and the tooth pitch deviation does not affect the gear meshing transmission performance. Moreover, in the actual production and processing process, the error measured by the CMM is the tooth form error of the tooth surface, so only the tooth form error generated by the heat treatment deformation needs to be compensated and corrected.

When the cutting parameters are adjusted, the adjusted tooth surface will deviate from the designed tooth surface, that is, the tooth surface error will appear. As shown in Figure 6, $W_{1}$ is the theoretical tooth surface and $W_{1}{}^{\prime }$ is the adjusted tooth surface.

Rotate the adjusted tooth surface *W*_1_ by a certain angle Δφ so that its center node *M′* coincides with the center node *M* of the theoretical tooth surface *W*_1_ to eliminate the tooth pitch deviation between the two tooth surfaces, and Δφ is the tooth pitch deviation. After eliminating the tooth pitch deviation, the distance along the normal direction of any point on the theoretical tooth surface to the corresponding point on the adjusted tooth surface is the normal error λ of the adjusted tooth surface at that point. The normal error of each point on the tooth surface constitutes the tooth form error of the gear. The formula of λ [32,33] is as follows:(5)$$\lambda =\left[M_{R}\cdot r_{1}{}^{\prime }\left(\theta_{1},\varphi_{1},k^{\prime }\right)-r_{1}\left(\theta_{1},\varphi_{1},k\right)\right]\cdot n\left(\theta_{1},\varphi_{1},k\right)$$

In the formula, $n\left(\theta_{1},\varphi_{1},k\right)$ is the unit normal vector of any point on the theoretical tooth surface, and $M_{R}$ is the rotation matrix for eliminating the tooth pitch deviation 27. The expression is as follows.
(6)$$M_{R}=\left[\begin{array}{cccc} 1 & 0 & 0 & 0\\ 0 & \cos\Delta \varphi & \sin\Delta \varphi & 0\\ 0 & -\sin\Delta \varphi & \cos\Delta \varphi & 0\\ 0 & 0 & 0 & 1 \end{array}\right]$$

### 4.3. The Influence of Cutting Parameters on Tooth Form Structure

The theoretical tooth surface is rotated and projected over an axis section of the gear rotation axis to obtain its topological plane. Taking the tooth profile direction of the topological plane as the X axis and the tooth direction as the Y axis, the normal error value of the tooth surface corresponding to any point on the topological plane is the Z coordinate value of the point; then, the differential surface between the theoretical tooth surface and the adjusted tooth surface is formed. A total of 45 uniformly distributed discrete tooth surface points in nine rows and five columns are selected within the range of the whole tooth surface, and the normal error value λ of the corresponding points is solved to establish the tooth surface error topology diagram, as shown in Figure 7.

When adjusting the cutting parameters, the differential surface is used to study the change of the tooth form of the adjusted tooth surface relative to the theoretical tooth surface.

In the research process, the each cutting parameters were fine-tuned, with an angle parameter +0.1°, displacement parameter +0.1 mm, and dimensionless constant +0.01. Observe the change of tooth form error, and summarize the influence law of each tooth cutting processing parameter on the normal error λ at each point of tooth surface. In the spiral bevel gear drive, the concave surface of the pinion and the convex surface of the large wheel are the working faces. In order to facilitate the machining, only the pinion is used for reverse adjustment of tooth cutting. This paper takes the concave surface of the pinion as an example to study the reverse adjustment of tooth cutting work.

Figure 8 shows the tooth surface error topology diagram corresponding to the fine-tuning of each cutting parameters. Through the analysis of Figure 8, it can be concluded that the influences of the adjustment of cutting parameters on the tooth form error are as follows:

The adjustment of tooth cutting parameters has little influence on the middle of the tooth surface, but has great influence on the heel, toe, tooth root and tooth tip of the tooth surface. The maximum value of tooth form error caused by the adjustment of cutting parameters is mainly concentrated at the opposite corner of the tooth surface and at the edge of the tooth surface.The influence of each cutting parameter on tooth form error is different. Among the 11 cutting parameters, the roll ratio has the greatest impact on the tooth form error, resulting in the cumulative tooth profile error of the tooth surface: 5.901 mm, followed by cutter radial setting, blade tilt angle, vertical offset, machine root angle and pressure angle of the blade, the cumulative tooth profile error was caused by 0.367 mm, 0.274 mm, 0.206 mm, 0.148 mm, 0.112 mm, respectively. Other cutting parameters have little influence on tooth form error.The influence of each cutting parameter on the tooth form error is different. The maximum positive value of tooth form error caused by fine adjustment of roll ratio is 0.2952 mm at the tip of the toe end, and the maximum negative value is 0.3584 mm at the root of the heel end. The roll ratio has great influence on the tooth profile and tooth direction error. The influence of vertical offset, cutter radial setting, sliding base feed setting and radius of the blade point on tooth form error mainly focuses on tooth direction, in which the vertical offset and cutter radial setting have greater effects, while the sliding base feed setting and radius of the blade point have fewer effects. The influence of machine root angle and pressure angle of the blade on tooth form error is mainly concentrated in the direction of tooth profile. The influence of blade tilt angle, blade swivel angle, basic cradle angle and increment of machine center to back on tooth form error is diagonally distributed.

### 4.4. Influence Weight Analysis of Cutting Parameters on Each Order Error of Tooth Form

In order to further investigate the specific impact of each cutting parameter on the errors of each order of tooth form, a second-order surface is used to approximate the tooth surface error topological surface. The expression of the second-order surface is:(7)$$Z=a_{1}X+a_{2}Y+a_{3}X^{2}+a_{4}Y^{2}+a_{5}XY$$
$a_{1}$, $a_{2}$ are the first-order error coefficient, representing the inclination of the differential surface on the *X* and *Y* axes, $a_{1}$ is the influence coefficient of pressure angle error and $a_{2}$ is the influence coefficient of spiral angle error. $a_{3}$, $a_{4}$, $a_{5}$ are the second-order error coefficients, representing the bending and distortion of the error surface, $a_{3}$ is the influence coefficient of the curvature error of the tooth profile, $a_{4}$ is the influence coefficient of the curvature error of the tooth direction and $a_{5}$ is the influence coefficient of the deflection of the tooth surface. The normal error of corresponding discrete points on the tooth surface is expressed by the matrix as:(8)$$\left(\begin{array}{c} Z_{1}\\ Z_{2}\\ Z_{3}\\ \vdots \\ Z_{45} \end{array}\right)=\left(\begin{array}{ccccc} X_{1} & Y_{1} & X_{1}^{2} & Y_{1}^{2} & X_{1}Y_{1}\\ X_{2} & Y_{2} & X_{2}^{2} & Y_{2}^{2} & X_{2}Y_{2}\\ X_{3} & Y_{3} & X_{3}^{2} & Y_{3}^{2} & X_{3}Y_{3}\\ \vdots & \vdots & \vdots & \vdots & \vdots \\ X_{45} & Y_{45} & X_{45}^{2} & Y_{45}^{2} & X_{45}Y_{45} \end{array}\right)\left(\begin{array}{c} a_{1}\\ a_{2}\\ a_{3}\\ a_{4}\\ a_{5} \end{array}\right)$$

According to the coordinates of the discrete points on the tooth surface and the normal error of the corresponding points, the error influence coefficient of each order is calculated by Formula (8), as shown in Table 4.

The larger the error influence coefficient, the greater the influence of cutting parameters on the tooth form error of that order. Analyzing Table 4 can obtain the influence weights of each cutting parameters on the tooth form error of each order:

From the comparison of each cutting parameter, the fine adjustment amount of the roll ratio is 0.01, and the fine adjustment amount of other parameters is 0.1. Therefore, the roll ratio has the greatest impact on the errors of each order of the tooth form. From the comparison of the influence coefficients of each order, it can be seen that the adjustment of tooth cutting parameters has a greater impact on the first-order error and a smaller impact on the second-order error.In addition to the roll ratio, the parameters that have a great influence on the pressure angle error are, successively: cutter radial setting > blade tilt angle > increment of machine center to back > radius of the blade point > vertical offset. The parameters that have great influence on the spiral angle error are, successively: blade tilt angle > pressure angle of the blade > machine root angle > vertical offset > cutter radial setting. The parameters that have great influence on the curvature error of tooth profile are, successively: cutter radial setting > machine root angle > increment of machine center to back > radius of the blade point > Sliding base feed setting. The parameters that have great influence on the curvature error of tooth direction are, successively: cutter radial setting > vertical offset > increment of machine center to back > Machine root angle > Blade tilt angle. The parameters that have great influence on the deflection error of tooth surface are, successively: cutter radial setting > Machine root angle > Blade tilt angle > vertical offset > Pressure angle of the blade.

## 5. Establishment and Experimental Verification of the Reverse Adjustment Correction Model of Tooth Cutting

The adjustment of each cutting parameter has a regular impact on the tooth form change of spiral bevel gears. Therefore, it is feasible to compensate for the tooth form deformation in heat treatment by adjusting the cutting parameters in reverse. Based on the established relationship between the adjustment of cutting parameters and the variation of tooth form error, a sensitivity coefficient matrix for tooth form error is constructed, and the reverse adjustment correction model of tooth cutting for heat treatment tooth form deformation of spiral bevel gear is established. The Levenberg–Marquardat method was used to solve the nonlinear least squares problem in the calculation of the parametric reverse adjustment amount of tooth cutting.

### 5.1. Establishment of the Reverse Adjustment Correction Model of Tooth Cutting

By partial derivation of the tooth form error of Formula (5) to the cutting parameters, we can obtain:(9)$$\frac{\partial \lambda }{\partial k}=-\frac{\partial r(\theta_{1},\varphi_{1},k)}{\partial k}\cdot n(\theta_{1},\varphi_{1},k)$$

The sensitivity coefficient of the *i*-th tooth cutting parameter to the tooth form error of the *j*-th discrete point on the tooth surface is denoted as $\eta_{j}^{i}$ [34], and the formula is as follows:(10)$$\eta_{j}^{i}=\frac{\partial \lambda_{j}}{\partial k_{i}}=-\frac{\partial r_{j}}{\partial k_{i}}\cdot n$$

The tooth form error of a certain point on the tooth surface can be considered as the superposition of the tooth form error caused by the adjustment of each cutting machining parameter Δk at this point. The tooth form error of a certain tooth surface caused by the adjustment of each cutting parameter can be expressed as:(11)$$\Delta \lambda_{j}=\eta_{j}^{1}\cdot \Delta k_{1}+\eta_{j}^{2}\cdot \Delta k_{2}+\cdots \cdots +\eta_{j}^{n}\cdot \Delta k_{n}=\sum_{i=1}^{n}\eta_{j}^{i}\cdot \Delta k_{i}$$

According to the measuring rules of spiral bevel gear, 45 uniformly distributed tooth surface measuring points were selected in nine rows and five columns in the tooth surface measuring area. The measurement point of tooth surface 45 and its corresponding sensitivity coefficients of each cutting parameter are put into Formula (11), and the tooth form error can be expressed in matrix form as:(12)$$\left[\begin{array}{c} \Delta \lambda_{1}\\ \Delta \lambda_{2}\\ \vdots \\ \Delta \lambda_{45} \end{array}\right]=\left[\begin{array}{cccc} \eta_{1}^{1} & \eta_{1}^{2} & \cdots & \eta_{1}^{n}\\ \eta_{2}^{1} & \eta_{2}^{2} & \cdots & \eta_{2}^{n}\\ \vdots & \vdots & \ddots & \vdots \\ \eta_{45}^{1} & \eta_{45}^{2} & \cdots & \eta_{45}^{n} \end{array}\right]\left[\begin{array}{c} \Delta k_{1}\\ \Delta k_{2}\\ \vdots \\ \Delta k_{n} \end{array}\right]$$

This can be expressed as: $\left[\Delta \lambda_{j}\right]=\left[\eta_{j}^{i}\right]\left[\Delta k_{i}\right]$ (*i* = 1∼*n*, *j* = 1∼45).

In the formula, *n* represents the number of cutting parameters, $\left[\Delta \lambda_{j}\right]$ represents the normal error value of each measurement point, and when in reverse adjustment of tooth cutting, it is the reserved adjustment allowance for each measurement point, which is equal to the deformation amount of the heat-treated tooth form, and the direction is opposite to the deformation direction of the heat treatment; $\left[\eta_{j}^{i}\right]$ is the sensitivity coefficient of tooth form error of each cutting parameter to each measurement point on the tooth surface; $\left[\Delta k_{i}\right]$ is the reverse adjustment amount of the cutting parameters. The number of cutting parameters in the tooth form error correction model is much less than the number of discrete points on the tooth surface, which is n≪45, so this equation system has transcendental stability. The Equations (12) are overdetermined nonlinear equations, in order to avoid the problem of numerical instability caused by the ill-conditioned or singular Jacobian matrix caused by nonlinear characteristics during solution [35]. Therefore, the solution of the reverse adjustment amount of tooth cutting can be transformed into a nonlinear least squares problem, and Formula (5) can be expressed as:(13)$$k=\arg\min\frac{1}{2}\lambda^{T}(k)\lambda (k)$$

The Levenberg–Marquardat method is used to solve the nonlinear least squares problem and obtain the reverse adjustment amount of cutting parameter. The basic idea of the Levenberg–Marquardat method [36,37] is to solve at each iteration:(14)$$\Delta k_{m+1}=\Delta k_{m}-{\left(\eta_{m}^{T}\eta_{m}+\mu_{m}I\right)}^{-1}\eta_{m}^{T}F_{m}\;\mu_{m}⩾0$$

In the formula, $\Delta k_{m+1}$ and $\Delta k_{m}$ are the reverse adjustment amount obtained by the *m* + 1 iteration and the *m* iteration, respectively; *F_m_* is the objective function value of iteration *m* (residual) and $\eta_{m}$ is the sensitivity coefficient matrix of the m-th iteration. $\mu_{m}$ is to overcome the singular nonnegative parameter $\eta_{m}^{T}\eta_{m}$.

### 5.2. Analysis of the Results of Reverse Tooth Cutting

For the heat-treated spiral bevel pinion gear mentioned above, import the tooth surface 45 measurement point into the DEFORM software, as shown in Figure 9.

Use the software point tracking function to measure the displacement changes of each measurement point after heat treatment, and obtain the tooth surface error after heat treatment. After the tooth pitch deviation is eliminated, the tooth form error in heat treatment of concave surface can be obtained as shown in Figure 10a, and the maximum tooth form error is −27.5 μm; the cumulative tooth form error is 425.3 μm.

After establishing the deformation of the tooth form in heat treatment, reverse adjustment of the cutting parameters is performed before heat treatment to compensate for the tooth deformation. Tooth surface reservation machining allowance is equal to the amount of thermal deformation theoretically, and the direction is opposite. In order to improve the efficiency of reverse adjustment of tooth cutting, select the tooth cutting parameters that have a significant impact on the tooth form errors of each order for reverse adjustment, and take into account the correction of the tooth form errors of each order. The selected cutting parameters of reverse adjustment are: roll ratio, pressure angle of the blade, vertical offset, cutter radial setting, blade tilt angle and machine root angle. According to the Formula (12) of the reverse adjustment correction model of tooth cutting, the reverse adjustment results of the cutting parameters can be calculated, as shown in Table 5.

After reverse adjustment of tooth cutting, heat treatment simulation was conducted, and the tooth form error after heat treatment is shown in Figure 10b. After the reverse adjustment, the maximum tooth form error is −9.18 μm, which is reduced by 66.62% compared with that before the reverse adjustment. The cumulative tooth form error is 185 μm, which is reduced by 56.5% compared with that before the reverse adjustment; the tooth form errors are effectively controlled.

### 5.3. Experimental Verification

In order to verify the effectiveness of the reverse adjustment correction model of tooth cutting in practical machining, the pinion is used as the experimental object. Firstly, gear cutting is performed on the spiral bevel gear [38]. After gear cutting, the gears are heat-treated, followed by heating, carburizing, quenching and tempering in the Aixielin ring furnace production line. The quenching method is direct quenching, and the quenching oil is Houghton-G model. After heat treatment, the tooth form error of the heat treatment is measured by using the Klingelnberg fully automatic CNC gear detection center, as shown in Figure 11.

The tooth form error of gear heat treatment is shown in Figure 12a. After reverse adjustment correction of tooth cutting, the gear is subjected to heat treatment processing, and the tooth form heat treatment deformation error after reverse adjustment correction is measured as shown in Figure 12b.

Comparing the experimental and simulation results, it can be seen that the tooth form deformation law obtained from the multi-field coupled heat treatment simulation is basically consistent with the experimental results, which verifies the accuracy of the heat treatment simulation. Before reverse adjustment, the maximum tooth form error of the tooth surface after heat treatment is −34.3 μm and the cumulative tooth form error is 618.8 μm. After reverse adjustment, the maximum tooth form error of the tooth surface is −8.7 μm, reduced by 74.64% compared with that before reverse adjustment, and the tooth form errors are all less than 10 μm after reverse adjustment, which meets the requirement of tooth form accuracy after heat treatment; the cumulative tooth form error is 199.8 μm, reduced by 67.71% compared with that before reverse adjustment, as shown in Figure 13.

From the comparison of tooth form errors after heat treatment before and after reverse adjustment of tooth cutting, it can be seen that the maximum and cumulative errors of tooth form have been significantly reduced after reverse adjustment of tooth cutting, and the deformation of heat-treated tooth form has been significantly reduced, verifying the effectiveness of the reverse adjustment correction model of tooth cutting for tooth form deformation in heat treatment of spiral bevel gear.

## 6. Discussion

In order to take into account the correction of tooth form errors of each order, the tooth cutting parameters which have great influence on tooth form errors of each order are selected for reverse adjustment, which are: roll ratio, pressure angle of the blade, vertical offset, cutter radial setting, blade tilt angle and machine root angle. It should be mentioned that the effect of the roll ratio on the tooth shape is much greater than that of other gear cutting parameters, and the roll ratio can be used as an important parameter in the tooth cutting process of spiral bevel gears.

The obtained heat treatment tooth profile error value reflects the size of the heat treatment tooth profile deformation. By comparing the tooth form errors of spiral bevel gears before and after tooth cutting reverse adjustment, we can see the effect before and after tooth cutting reverse adjustment. In order to verify the reverse adjustment correction model of tooth cutting for heat treatment tooth form deformation of spiral bevel gear, heat treatment simulation and gear processing and heat treatment experiments were carried out, respectively. Comparing the simulation results and the experimental results of the tooth form error before the tooth cutting reverse adjustment, the maximum tooth form error value obtained by the simulation is −27.5 μm, and the maximum tooth form error value obtained by the experiment is −34.3 μm. The cumulative tooth form error value obtained by simulation is 425.3 μm; the cumulative tooth form error value obtained by experiment is 618.8 μm. Obviously, the experimental value is greater than the simulation value, which may be caused by the influence of gear material impurities in the actual heat treatment processing, the way of loading gear in the heat treatment and the release of machining stress during machining. However, heat treatment simulation cannot take these factors into account.

In some cases, there are not only tooth form errors caused by heat treatment tooth profile deformation, but also tooth form errors caused by cutting, machine tool errors and gear pair installation errors during processing, which have reference significance for the reverse correction of these errors. In addition, the reverse adjustment correction model of tooth cutting proposed in this paper is also applicable to the gear-grinding process.

## 7. Conclusions

Aiming at the problem of tooth form deformation in heat treatment of spiral bevel gears, this paper thoroughly researches the influence law of cutting parameters on tooth form error and further establishes the reverse adjustment correction model of tooth cutting for heat treatment tooth form deformation of spiral bevel gear. By adjusting the cutting parameters and reserving machining allowance during the cutting stage, it compensates for the deformation of the heat-treated tooth form of spiral bevel gears and realizes the control of heat treatment tooth form deformation. The main conclusions are as follows:

Based on the cutting parameters, a mathematical model of the tooth surface of spiral bevel gears was established, and the heat treatment process of the spiral bevel gear was simulated by multi-field coupling simulation. The results show that: after heat treatment, the depth of the carburized layer on the tooth surface is 1.05 mm, the carbon content on the tooth surface is 0.764 and the distribution of carbon elements is uniform. The hardness of the tooth surface is 62.8HRC, and the hardness of the gear core is 33HRC, which meets the performance requirements for heat treatment processing. The research method of using multi-field coupled simulation can reduce the cost and improve the efficiency of heat treatment research.The influence law of each cutting parameter on the concave tooth form error of the pinion was investigated by using the method of artificially given cutting parameters with small disturbance variables. Among each cutting parameter, the roll ratio has the greatest impact on tooth form error; The maximum tooth form error of the pinion concave surface caused by fine-tuning the roll ratio by 0.01 is −0.3584 mm, and the cumulative tooth form error is 5.901 mm. Followed by cutter radial setting, blade tilt angle, vertical offset, machine root angle and pressure angle of the blade, the above cutting parameters can be used as an important item of cutting adjustment.In this paper, we propose a reverse adjustment correction model of tooth cutting for heat treatment tooth form deformation of spiral bevel gear, based on the Levenberg–Marquardat method to solve the stable and accurate numerical solution for the reverse adjustment amount of the cutting parameters. In addition, an experimental study of reverse adjustment of tooth cutting processing was carried out, and the experimental results showed that after the reverse gear cutting, the maximum tooth form error of the heat treatment deformation on the concave surface of the pinion is 8.7 μm, reduced by 74.64% compared to that before the reverse gear cutting. The cumulative tooth form error is 199.8 μm, reduced by 67.71% compared to that before the reverse gear cutting. The validity of the reverse adjustment correction model of tooth cutting was verified. The research in this paper can provide a new option for heat treatment tooth form deformation control of spiral bevel gears. The reverse adjustment correction model of tooth cutting proposed in this paper and the research on the influence of cutting parameters on tooth form can provide a technical support and theoretical reference for high-precision tooth cutting of spiral bevel gear.

In this paper, only the tooth surface modeling and tooth cutting reverse adjustment research are carried out for the spiral bevel gear processed by the tool tilt method. This processing method is more complicated and the processing efficiency is low. It is necessary to study the tooth surface modeling and tooth cutting reverse adjustment of new processing methods, such as the double helix method. In this paper, the Levenberg–Marquardat method is used to calculate the reverse of tooth cutting parameters, which needs to be further improved.

## Figures and Tables

**Figure 1 materials-16-04183-f001:**
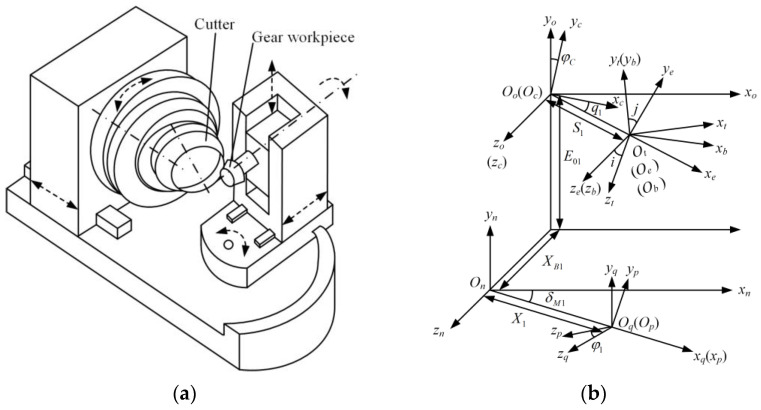
Machine Tool Structure and Coordinate System. (**a**) Machine tool structure diagram; (**b**) Coordinate diagram of machine tool and workpiece.

**Figure 2 materials-16-04183-f002:**
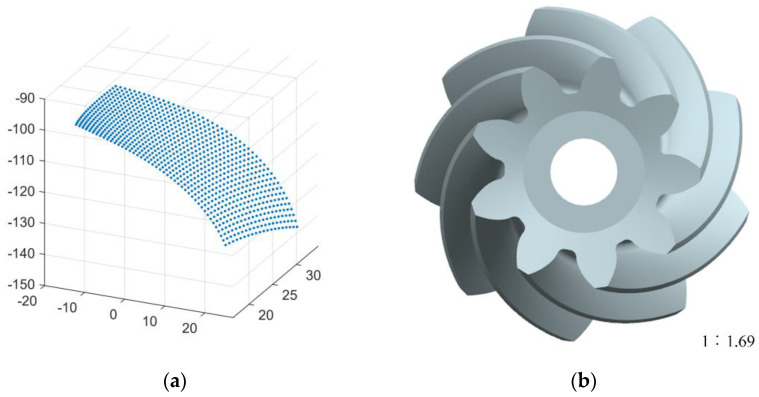
Discrete points of tooth surface and 3D model of pinion. (**a**) Discrete points of tooth surface; (**b**) 3D model of pinion.

**Figure 3 materials-16-04183-f003:**
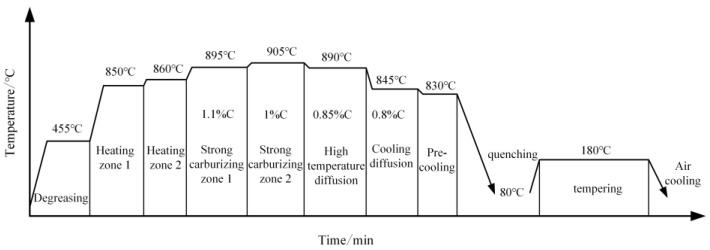
Heat treatment process flow.

**Figure 4 materials-16-04183-f004:**
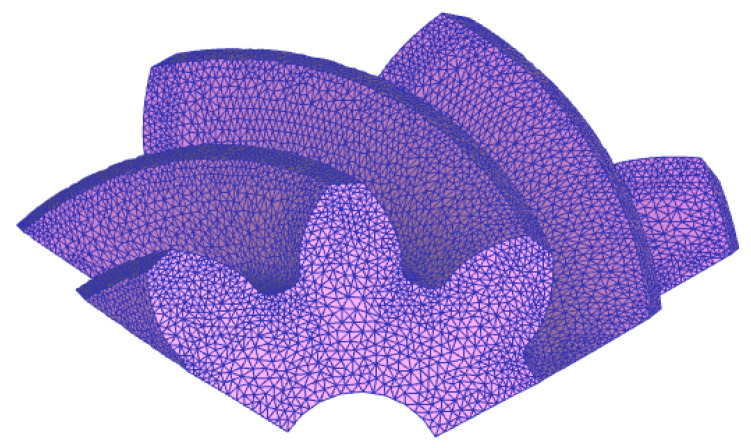
Finite element mesh model.

**Figure 5 materials-16-04183-f005:**
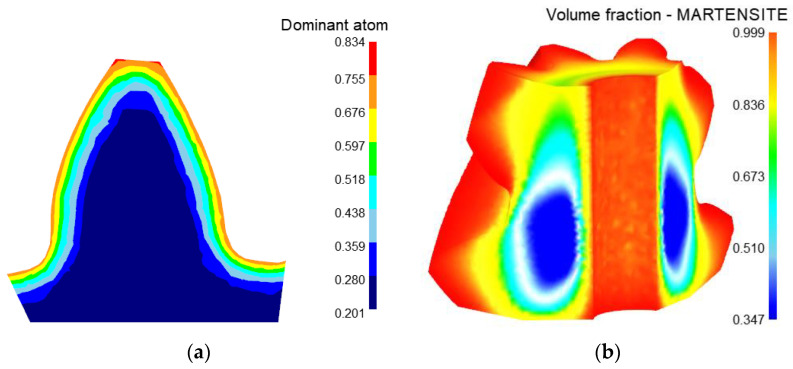
Heat treatment simulation results. (**a**) Distribution of carbon elements; (**b**) Distribution of martensite; (**c**) Distribution of hardness; (**d**) Deformation of tooth surface.

**Figure 6 materials-16-04183-f006:**
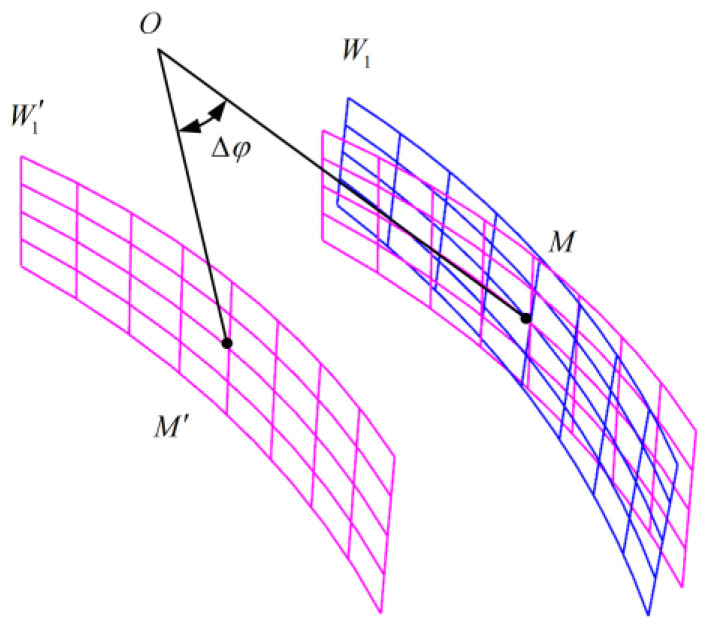
Elimination of tooth pitch deviation.

**Figure 7 materials-16-04183-f007:**
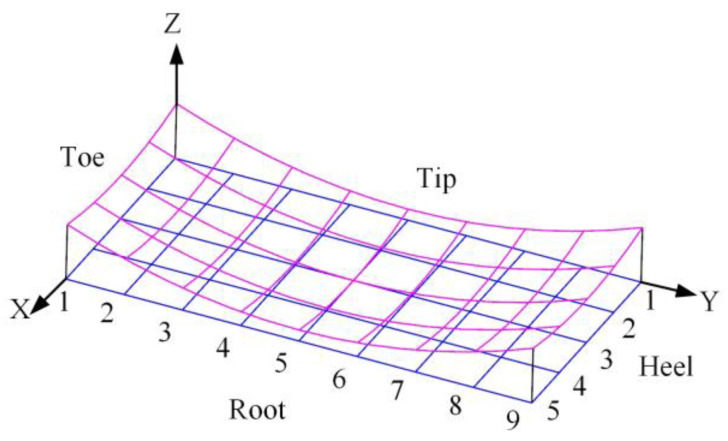
Topological diagram of tooth surface error.

**Figure 8 materials-16-04183-f008:**
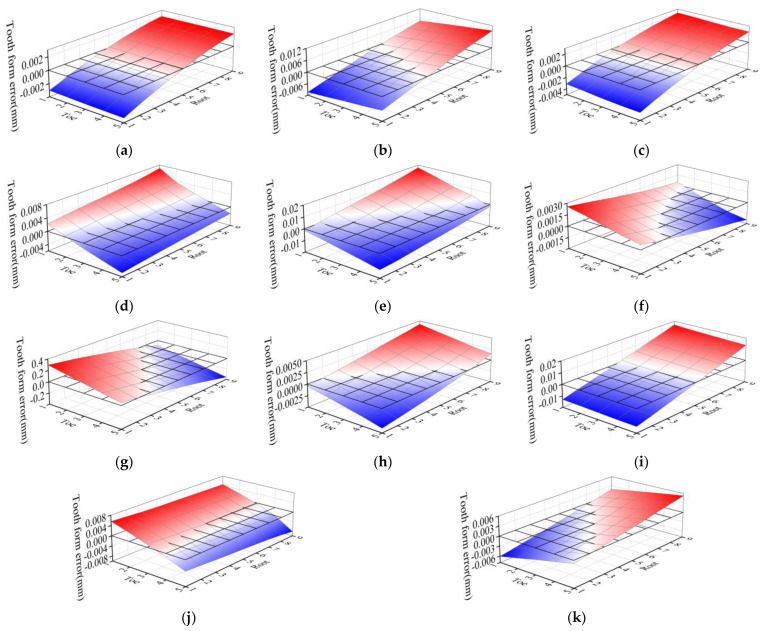
Influence of fine-tuning of cutting parameters on topological structure of tooth surface. (**a**) Sliding base feed setting +0.1 mm; (**b**) Vertical offset +0.1 mm; (**c**) Radius of the blade point +0.1 mm; (**d**) Pressure angle of the blade +0.1°; (**e**) Blade tilt angle +0.1°; (**f**) Blade swivel angle +0.1°; (**g**) Roll ratio +0.01; (**h**) Basic cradle angle +0.1°; (**i**) Cutter radial setting +0.1 mm; (**j**) Machine root angle +0.1°; (**k**) Increment of machine center to back +0.1 mm.

**Figure 9 materials-16-04183-f009:**
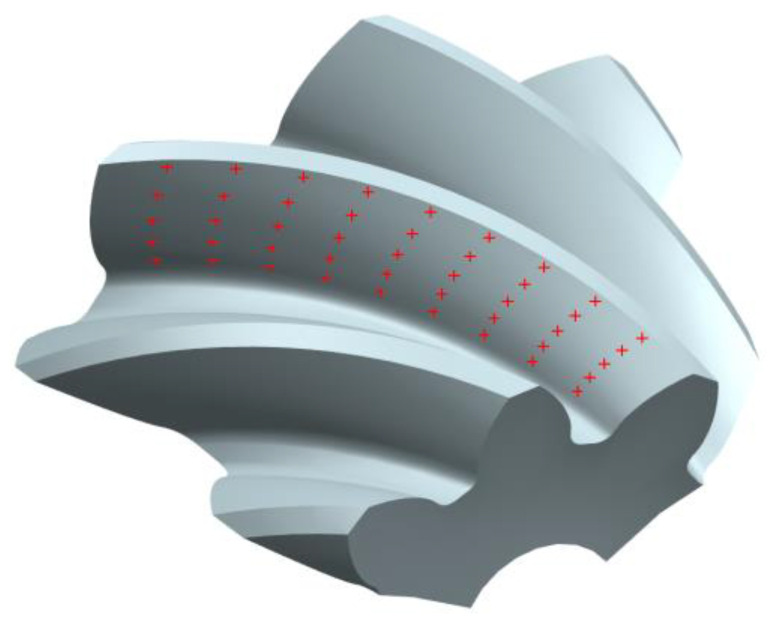
Measuring points of tooth surface.

**Figure 10 materials-16-04183-f010:**
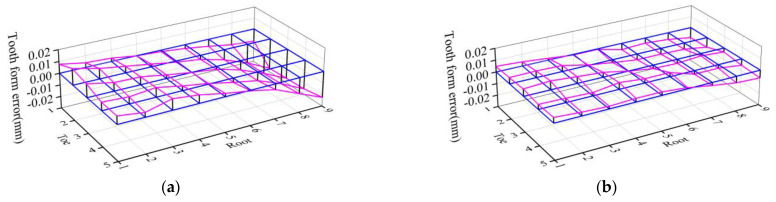
Tooth form error of heat treatment. (**a**) Tooth form error of heat treatment before reverse adjustment; (**b**) Tooth form error of heat treatment after reverse adjustment.

**Figure 11 materials-16-04183-f011:**
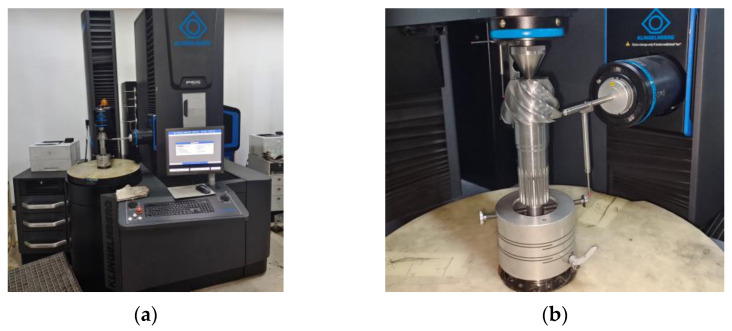
Measurement of tooth form error. (**a**) Klingelnberg fully automatic CNC gear detection center; (**b**) Gear and gauging probes.

**Figure 12 materials-16-04183-f012:**
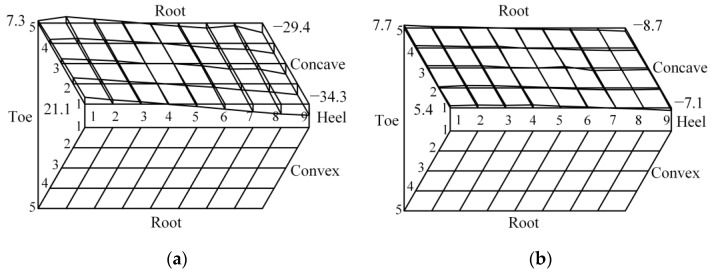
Concave tooth form error. (**a**) Tooth form error before reverse adjustment; (**b**) Tooth form error after reverse adjustment.

**Figure 13 materials-16-04183-f013:**
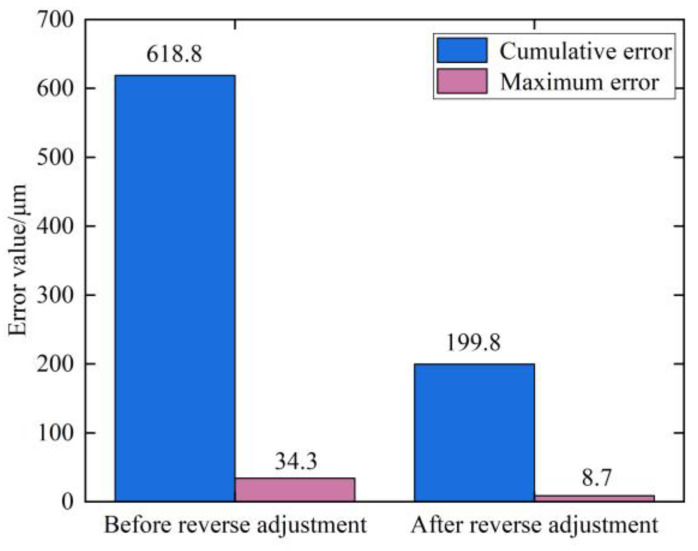
Statistics of accumulated error and maximum error of tooth form.

**Table 1 materials-16-04183-t001:** The basic design parameters of the pinion.

Gear Parameters	Pinion
Rotation	left
Number of tooth *Z*	9
Face width b/mm	45
$\mathrm{Mean}\;\mathrm{normal}\;\mathrm{module}\;m_{n}/\mathrm{mm}$	7.1795
$\mathrm{Pressure}\;\mathrm{angle}\;\alpha_{n}{/}^{\circ }$	20
$\mathrm{Spiral}\;\mathrm{angle}\;\beta {/}^{\circ }$	35
$\mathrm{Pitch}\;\mathrm{cone}\;\mathrm{distance}\;L_{a}/\mathrm{mm}$	136.412
$\mathrm{Pitch}\;\mathrm{angle}\;\delta {/}^{\circ }$	13
$\mathrm{Root}\;\mathrm{angle}\;\delta_{f}{/}^{\circ }$	11.67
$\mathrm{Face}\;\mathrm{angle}\;\delta_{a}{/}^{\circ }$	16.94

**Table 2 materials-16-04183-t002:** The tooth cutting parameters of the pinion.

The Parameters of Cutting	Concave	Convex
$\mathrm{Pressure}\;\mathrm{angle}\;\mathrm{of}\;\mathrm{the}\;\mathrm{blade}\;\alpha_{1}{/}^{\circ }$	14	31
$\mathrm{Radius}\;\mathrm{of}\;\mathrm{the}\;\mathrm{blade}\;\mathrm{point}\;r_{01}/\mathrm{mm}$	113.3	115.3
$\mathrm{Machine}\;\mathrm{root}\;\mathrm{angle}\;\delta_{M1}{/}^{\circ }$	13.367	13.5
$\mathrm{Increment}\;\mathrm{of}\;\mathrm{machine}\;\mathrm{center}\;\mathrm{to}\;\mathrm{back}\;X_{1}/\mathrm{mm}$	−4.75	5.1
$\mathrm{Vertical}\;\mathrm{offset}\;E_{01}/\mathrm{mm}$	0	0
$\mathrm{Sliding}\;\mathrm{base}\;\mathrm{feed}\;\mathrm{setting}\;X_{B1}/\mathrm{mm}$	7.99	15.37
$\mathrm{Basic}\;\mathrm{cradle}\;\mathrm{angle}\;q_{1}{/}^{\circ }$	106.709	92.126
$\mathrm{Cutter}\;\mathrm{radial}\;\mathrm{setting}\;S_{1}/\mathrm{mm}$	133.348	142.856
$\mathrm{Blade}\;\mathrm{tilt}\;\mathrm{angle}\;i{/}^{\circ }$	24.083	26.35
$\mathrm{Blade}\;\mathrm{swivel}\;\mathrm{angle}\;j{/}^{\circ }$	44.75	33.667
$\mathrm{Roll}\;\mathrm{ratio}\;i_{01}$	4.27972	4.63017

**Table 3 materials-16-04183-t003:** The main chemical composition of 20CrMnTi.

Composition	C	Si	Mn	Cr	Ti	S	P	Fe
Mass fraction (%)	0.2	0.21	1.02	1.23	0.04	0.014	0.023	97.263

**Table 4 materials-16-04183-t004:** Error influence coefficient of each stage on the concave surface of the pinion.

The Parameters of Tooth Cutting	$a_{1}$	$a_{2}$	$a_{3}$	$a_{4}$	$a_{5}$
Pressure angle of the blade	−0.0008019	0.0013203	−0.0000559	−0.0000435	−0.0001384
Radius of the blade point	−0.0019004	0.0004029	0.0002002	−0.0000031	0.0000910
Machine root angle	0.0017280	0.0010451	−0.0005273	−0.0000541	−0.0002137
Increment of machine center to back	−0.0020634	−0.0005740	0.0003825	0.0000951	0.0001293
Vertical offset	−0.0018171	−0.0007051	0.0001124	0.0001276	0.0001482
Sliding base feed setting	−0.0017299	0.0003203	0.0001863	0.0000006	0.0000908
Basic cradle angle	−0.0008762	0.0004222	−0.0000078	0.0000148	0.0000185
Cutter radial setting	−0.0072466	0.0006348	0.0007961	0.0001883	0.0002969
Blade tilt angle	−0.0025545	0.0020072	−0.0000667	0.0000463	−0.0001970
Blade swivel angle	0.0008358	0.0002963	−0.0001347	−0.0000281	−0.0001039
Roll ratio	0.1006654	0.0385447	−0.0185606	−0.0056669	−0.0087164

**Table 5 materials-16-04183-t005:** Reverse adjustment amount of tooth cutting parameters.

The Parameters of Cutting	Value before Reverse	Amount of Reverse Adjustment	Value after Reverse
$\mathrm{Pressure}\;\mathrm{angle}\;\mathrm{of}\;\mathrm{the}\;\mathrm{blade}\;\alpha_{1}{/}^{\circ }$	14	−0.21499	13.78501
$\mathrm{Machine}\;\mathrm{root}\;\mathrm{angle}\;\delta_{M1}{/}^{\circ }$	13.367	−0.16114	13.20586
$\mathrm{Vertical}\;\mathrm{offset}\;E_{01}/\mathrm{mm}$	0	−0.37099	−0.37099
$\mathrm{Cutter}\;\mathrm{radial}\;\mathrm{setting}\;S_{1}/\mathrm{mm}$	133.348	0.18277	133.53077
$\mathrm{Blade}\;\mathrm{tilt}\;\mathrm{angle}\;i{/}^{\circ }$	24.083	0.14494	24.22794
$\mathrm{Roll}\;\mathrm{ratio}\;i_{01}$	4.27972	−0.00021	4.27951

## Data Availability

Not applicable.

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
