# Peer review of "Control of Tooth Form Deformation in Heat Treatment of Spiral Bevel Gears Based on Reverse Adjustment of Cutting Parameters"

_materials, 2023, doi:10.3390/ma16114183_

Round 1

Reviewer 1 Report

The authors have conducted an impressive study on controlling tooth form deformation in heat treatment of spiral 2 bevel gears. The research methodology and results are clearly presented, and the conclusions drawn are sound and informative. The authors' innovative approach of reverse adjustment of cutting 3 parameters is particularly noteworthy, and the experimental verification of their findings is a valuable contribution to the field.

However, it should be noted that the absence of a discussion section is a limitation of this study. A discussion section would have provided an opportunity for the authors to interpret their findings, relate their results to previous research, and discuss the implications of their work for future studies.

Overall, I believe that this research paper makes an important contribution to the field of materials science and engineering. The findings presented are of significant interest to researchers and practitioners in this field, and the authors' work is deserving of publication in the Materials journal.

Overall, I found the paper to be well-written and informative. The authors provide a detailed description of their research methodology and the results they obtained. However, I did notice a few areas where the paper could be improved.

Firstly, I noticed that the paper does not have a limitation paragraph. It would be helpful for the authors to acknowledge any potential limitations or shortcomings of their research. This would provide readers with a more complete understanding of the study's scope and implications.

Secondly, I believe that the paper could benefit from a discussion section. This section would allow the authors to contextualize their findings within the broader literature and to explain the significance of their results in more detail. Without a discussion section, readers may be left wondering about the broader implications of the research.

Finally, I would encourage the authors to provide more explanation regarding the critical significance of their work. While they do briefly touch on this in the introduction, I believe that it would be helpful for them to expand on this point and to clearly articulate why their research is important and valuable.

Reviewer 2 Report

The paper “materials-2401703” related to machining was reviewed. Please follow the comments carefully and resubmit your paper for the next consideration and reviewing process.

1.     How did you select the process parameters in Table 1?

2.     Add the scale bar for Figure 2 (the cutting tool).

3.     Please add a brief statement on your methodology in the abstract.

4.     Add brief quantitative results to the abstract

5.      Subtractive (machining) and additive methods are comparable which can be highlighted in your paper. Please read the following manuscript and add it to the literature to show how the subtractive (machining) is comparable with other manufacturing. “Laser subtractive and laser powder bed fusion of metals: review of process and production features”

6.      List the detail of the findings in your conclusion.

Needs some works. 

Reviewer 3 Report

Paper shows a good procedure for increasing the quality of teeth of bevel gears after heat treatment. Paper has all elements of a scientific paper: numerical approuch, FEM results, whole procedure is given, results with changed geometry and experimental verification. I suggest this paper for publishing. 

"Reverse adjustment of cutting parameters" is not appropriate for keyword. Better is "reverse engineering" and "cutting parameter"

line 179 - Capital letter

Reviewer 4 Report

My comments are in the PDF attached.

Round 2

Reviewer 2 Report

It is ready to publish.

Reviewer 4 Report

Accepted.